# Ten-Year Assessment of IASI Radiance and Temperature

**Marie Bouillon** [1,*]**, Sarah Safieddine** [1] **, Juliette Hadji-Lazaro** [1]**, Simon Whitburn** [2]**,**
**Lieven Clarisse** [2]**, Marie Doutriaux-Boucher** [3]**, Dorothée Coppens** [3]**, Thomas August** [3]**,**
**Elsa Jacquette** [4] **and Cathy Clerbaux** [1,2]

1   LATMOS/IPSL, Sorbonne Université, UVSQ, CNRS, 75005 Paris, France;
    sarah.safieddine@latmos.ipsl.fr (S.S.); juliette.hadji-lazaro@latmos.ipsl.fr (J.H.-L.);
    cathy.clerbaux@latmos.ipsl.fr (C.C.)
2   Spectroscopy, Quantum Chemistry and Atmospheric Remote Sensing (SQUARES), Université Libre de
    Bruxelles (ULB), 1050 Brussels, Belgium; simon.whitburn@ulb.ac.be (S.W.); lieven.clarisse@ulb.ac.be (L.C.)
3   European Organisation for the Exploitation of Meteorological Satellites, 64295 Darmstadt, Germany;
    marie.doutriauxboucher@eumetsat.int (M.D.-B.); dorothee.coppens@eumetsat.int (D.C.);
    thomas.august@eumetsat.int (T.A.)
4   Centre National d'Etudes Spatiales, 31400 Toulouse, France; elsa.jacquette@cnes.fr
*   Correspondence: marie.bouillon@latmos.ipsl.fr

**Abstract:** The Infrared Atmospheric Sounding Interferometers (IASIs) are three instruments flying on board the Metop satellites, launched in 2006 (IASI-A), 2012 (IASI-B), and 2018 (IASI-C). They measure infrared radiance from the Earth and atmosphere system, from which the atmospheric composition and temperature can be retrieved using dedicated algorithms, forming the Level 2 (L2) product. The operational near real-time processing of IASI data is conducted by the EUropean organisation for the exploitation of METeorological SATellites (EUMETSAT). It has improved over time, but due to IASI's large data flow, the whole dataset has not yet been reprocessed backwards. A necessary step that must be completed before initiating this reprocessing is to uniformize the IASI radiance record (Level 1C), which has also changed with time due to various instrumental and software modifications. In 2019, EUMETSAT released a reprocessed IASI-A 2007–2017 radiance dataset that is consistent with both the L1C product generated after 2017 and with IASI-B. First, this study aimed to assess the changes in radiance associated with this update by comparing the operational and reprocessed datasets. The differences in the brightness temperature ranged from 0.02 K at 700 cm$^{-1}$ to 0.1 K at 2200 cm$^{-1}$. Additionally, two major updates in 2010 and 2013 were seen to have the largest impact. Then, we investigated the effects on the retrieved temperatures due to successive upgrades to the Level 2 processing chain. We compared IASI L2 with ERA5 reanalysis temperatures. We found differences of ~5–10 K at the surface and between 1 and 5 K in the atmosphere. These differences decreased abruptly after the release of the IASI L2 processor version 6 in 2014. These results suggest that it is not recommended to use the IASI inhomogeneous temperature products for trend analysis, both for temperature and trace gas trends.

**Keywords:** IASI; climate studies; radiance; atmospheric temperature; surface temperature; atmospheric composition

## 1. Introduction

Surface and atmospheric temperatures are both Essential Climate Variables (ECV) that critically contribute to the characterization of Earth's climate [1]. In the past few decades, significant warming at the surface and in the troposphere has been observed due to the increase of greenhouse gases [2–5]. The opposite trend has been observed in the stratosphere. Stratospheric temperatures are driven by

both anthropogenic forcings, such as the greenhouse gas and ozone-depleting substance concentration, and natural forcings, such as volcanic eruptions and the solar cycle [6]. Cooling has been observed in the lower stratosphere due to ozone depletion and an increase in the greenhouse gas concentration has been observed in the middle and upper stratosphere [7–9].

Surface and atmospheric temperatures are derived from various instrument measurements. The longest continuous and consistent records are obtained from radiosondes and ground-based Lidar. However, these methods of observation have a limited spatial coverage, and they are unevenly distributed around the globe, with little to no measurements over the oceans and poles. Recent efforts in the Global Climate Observing System (GCOS) Reference Upper-Air Network (GRUAN) [10] have aimed to standardize quality assurance practices in releasing sondes and processing their data at a few stations. However, overall, they consist of inhomogeneous records (e.g., from different instruments, calibration, and data processing). In contrast, more recently, satellite-derived temperatures have become available, providing frequent and global observations of the atmosphere, and have thus become a key component for climate change monitoring [11,12]. To construct a long temperature time-series from satellite measurements, compiling and averaging data from several instruments is necessary. For this, adjustments and bias corrections between the different instruments are required [13,14]. Other biases are also related to the diversity in instrument characteristics and temperature retrieval algorithms that are based on different assumptions. When these datasets are used in climate and Numerical Weather Prediction (NWP) models, the error due to this homogenization of different data records becomes difficult to assess. This emphasizes the importance of using a single instrument with a global spatial coverage and long time series that is homogeneous and consistent for the assessment of climate variables.

Since 2006, the Infrared Atmospheric Sounding Interferometers (IASIs) have been used for numerical weather prediction [15] and to monitor the atmospheric composition [16,17]. IASI is also used as a reference for the inter-calibration of infrared sensors by the Global Space-Based Inter-Calibration System [18]. Inter-comparison of the three instruments has shown excellent agreement between them [19,20].

IASI radiance and temperature values, called Level 1C data (L1C) and Level 2 data (L2), respectively, are processed and delivered to numerous users by the Eumetcast delivery system. Since 2007, the EUropean organisation for the exploitation of METeorological SATellites (EUMETSAT) has carried out several updates of the processing algorithms for both L1C and L2 datasets and, as a result, the radiance and temperature time series are not homogeneous. Trace gas concentration retrievals that are based on IASI L1C and L2 are impacted by this non-homogeneity [21,22], making the computation of trends for temperature and atmospheric composition difficult.

Recently, EUMETSAT has reprocessed the 2007–2017 IASI Metop-A L1C data [23] with the most recent version of the algorithm and there is now a homogeneous L1C dataset available (data after 2017 are assumed to be homogeneous with the new dataset). EUMETSAT is now in the process of releasing a first homogeneous L2 temperature data record, but this is not yet available. This study thus aims to investigate the changes of L1C due to algorithm updates by comparing the operational IASI-A L1C data (non-homogeneous) with the reprocessed IASI-A L1C product (homogeneous). The inhomogeneity in the L2 records due to successive updates of the L2 algorithm is then evaluated by comparing the IASI temperatures at different altitudes with those from the European Centre for Medium-Range Weather Forecasts' (ECMWF) latest available reanalysis data—ERA5 [24].

## 2. Materials and Methods

### 2.1. The IASI Instrument

IASI is a Fourier transform spectrometer that measures the thermal infrared radiation emitted by the Earth and the atmosphere [16]. Three IASI instruments are currently operational on board the Metop satellites: IASI-A was launched in 2006, IASI-B in 2012, and IASI-C in 2018. The three IASI

instruments fly at an altitude of 817 km in a sun-synchronous orbit (98.7° inclination) and they observe the Earth with a swath width of 2200 km on the ground. Each swath contains 30 fields of view (15 on each side of the nadir) and each field of view is made up of four pixels that have a diameter of 12 km at each nadir. This observation mode allows each IASI instrument to fly over every location on Earth twice a day, around 09:30 and 21:30 local time.

For each pixel, IASI measures a radiance spectrum composed of 8461 channels between 645 and 2760 cm$^{-1}$ (between 3.62 and 15.5 µm), with a spectral resolution of 0.5 cm$^{-1}$ (apodized). Each of the three IASI measures about 1.3 million spectra per day.

### 2.1.1. IASI L1C Radiances

The radiance values are written as an integer N multiplied by a scale factor (N $\times$ 10$^{-\text{scalefactor}}$). EUMETSAT has divided the IASI spectrum into five spectral regions and each region has a scale factor that is approximately proportional to the value of the radiance values in the region. Figure 1 shows the precision of the radiance values as a function of the wavenumber. We can clearly see that the decimal precision of the radiance values depends on the wavenumber and varies between 10$^{-9}$ and 10$^{-7}$.

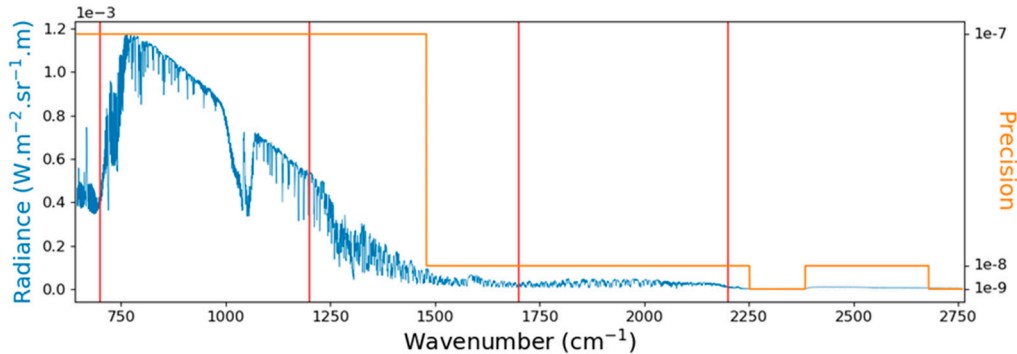

**Figure 1.** The Infrared Atmospheric Sounding Interferometer (IASI) typical radiance spectrum (left axis) superimposed with the precision used by the EUropean organisation for the exploitation of METeorological SATellites (EUMETSAT) (right axis), making the precision of the radiance values dependent on the wavenumber. The vertical red lines are the wavenumbers at which the differences were computed.

To compare the reprocessed IASI-A radiance values and the operational ones (before reprocessing), we looked for one coincident observation per day in each dataset. Slight differences (of the order of 10$^{-2}$ degrees) were found when comparing the latitudes and longitudes of coincident observations. For a proper comparison, the selected observations were the ones minimizing $(lat_{oper} - lat_{reproc})^2 + (lon_{oper} - lon_{reproc})^2$. By minimizing the difference between the operational and reprocessed latitudes and longitudes, we made sure that the observations selected in the operational and reprocessed datasets were the same.

The differences between the reprocessed and operational radiances were computed at four distinct wavenumbers (700, 1200, 1700, and 2200 cm$^{-1}$). Those wavenumbers were chosen because they are relevant in the study of atmospheric temperatures and humidity. The 700 and 2200 cm$^{-1}$ spectral channels were used to retrieve temperature profiles from the $CO_2$ bands. The 1200 cm$^{-1}$ channel is sensitive to the skin temperature and the 1700 cm$^{-1}$ band falls in the middle of the water vapor continuum [16,25].

Table 1 lists the main updates of the L1C algorithm. The updates are due to an update of a parameter used as an input in the L1 Product Processing Facility (PPF) (noted as "param") or by algorithm evolution inside the L1 PPF processor ("algo"). The IASI L1 PPF software version indicated in the table below is the version used for the operational IASI-A products (before reprocessing), and it is given for information, but is not linked to the change of parameters. The first reprocessed release

consists of one data record generated by version 8.0 of the EUMETSAT operational IASI processing chain and the most recent updated auxiliary files provided by the Centre National d'Etudes Spatiales (CNES), using the full orbit level 0 products retrieved from the EUMETSAT archive.

**Table 1.** List of IASI-A L1 processing updates.

| Date | L1 Update | IASI L1 PPF Software Version |
|---|---|---|
| 19 February 2009 | Routine on-board radiometric calibration update | |
| 12 May 2009 | Routine scan mirror reflectivity update (param) | |
| 18 May 2010 | "Day-2 evolution" improvement of the spectral calibration (param+algo) | 5.0.2 |
| 7 February 2011 | Update of the spectral harmonization functions and routine scan mirror reflectivity update (param) → pixel 2 in line with the other pixels | 5.1 |
| 20 April 2011 | Routine on-board radiometric calibration update | |
| 18 July 2012 | Routine scan mirror reflectivity update (param) | |
| 16 May 2013 | Change of the Instrument Point Spread Function (IPSF) and internal geometrical offset between the sounder and the integrated imager (param) | 6.2 |
| 28 August 2013 | Routine on-board radiometric calibration update | 6.5 |
| 16 September 2013 | Routine scan mirror reflectivity update (param) | |
| 17 June 2014 | New change of the internal geometrical offset between the sounder and the integrated imager (param) | 7.0 |
| 24 June 2015 | Routine scan mirror reflectivity update (param) | |
| 5 August 2015 | Routine on-board radiometric calibration update (param) | 7.3 |
| 7 February 2017 | Routine scan mirror reflectivity update (param) | 7.4 |

### 2.1.2. IASI L2 Temperatures

In the L2 operational product, IASI surface temperatures and profiles are computed for each IASI spectrum. There are three versions of the operational software used to retrieve IASI temperature profiles: from November 2007 to September 2010 (version 4), from September 2010 to September 2014 (version 5), and after September 2014 (version 6) [26,27]. Since version 6, IASI temperatures have been computed with the help of observations from two microwave instruments also onboard the Metop satellites: The Advanced Microwave Sounding Unit (AMSU) and the Microwave Humidity Sounder (MHS).

In all versions, statistical retrieval is performed first. It is then used as a first guess for variational retrieval, by implementing the optimal estimation method (OEM) [28], where it is further refined. While the statistical retrieval can be applied in nearly all sky conditions, the OEM is only attempted in cloud-free conditions.

In this approach, IASI first-guess profiles are incorporated into a fast radiative transfer model to compute synthetic radiance spectra and Jacobians. The first-guess profiles are then iteratively improved to minimize the following cost function:

$$J = (x - x_a)^T \, S_x^{-1}(x - x_a) + (F(x) - y)^T \, S_y^{-1}(F(x) - y), \tag{1}$$

where $x$ is the state vector, $x_a$ is the a priori knowledge of the state vector and $S_x$ is the associated covariance, $y$ is the observation vector, $F(x)$ is the forward model, and $S_y$ is the error covariance.

The difference between versions 4, 5, and 6 essentially reside in the configuration of the optimal estimation (channel selection, background, and observation error matrices), the radiative transfer model, the cloud mask, and the statistical method itself. A summary of the main evolution is presented hereafter. From version 5 onwards, the channel selection was designed to maximize the information content in the OEM through the application of principal components (PC) analysis, as well as to reduce the instrument noise. The radiative transfer was initially based on Radiative Transfer for IASI (RTIASI) [29] and implemented successive revisions of Radiative Transfer for TOVS (RTIASI) [30], until version 12. The background term was based on a static climatological average until version 5, and it has been an a priori variable since version 6, using the statistical inference of the first step. The background error is defined as the covariance of the differences between the first guess and ECMWF analysis. The observation error since version 5 has been the covariance of the differences

between the observed and simulated IASI radiance values, using the first guess as the input to RTTOV. Both are computed from a statistically large matchup (IASI observations and retrievals, ECMWF analysis) dataset. The basic principle of the first statistical retrieval is a linear regression between IASI observations and the atmospheric state vector, computed from a large and representative training set. Version 6 implements a higher level of sophistication: The linear relationship between IASI observations and geophysical parameters is searched in parts, in different observation classes resulting from k-mean unsupervised clustering. It also exploits adjacent pixels to take advantage of geophysical horizontal correlations. This forms the Piece-Wise Linear Regression-cube (PWLR$^3$) algorithm, whose purpose is to ensure that a linear relationship is a good approximation of the actual relationship between the predictors (IASI radiance values in PCs) and the predictands (e.g., atmospheric profiles in PCs) [31].

The temperature profiles used in this study are a combination of retrievals obtained with the statistical method and the optimal estimation method (OEM), which uses the statistical method as a first guess. The OEM is only applicable in cloud-free scenes and the dataset is hence completed by statistical retrievals in cloudy pixels. The record is still heterogeneous at this stage, as a systematic reprocessing of the IASI L2 products has not yet been released. It is therefore composed of successive operational versions of the IASI L2 processor, including incremental algorithm improvements. The exploitation of microwave radiance values in synergy with IASI started at the end of September 2014, with the version 6 IASI L2 processor, which significantly improved the product yield and quality.

Version 4 and 5 temperature profiles are given at 90 pressure levels and version 6 temperature profiles are given at 101 pressure levels, following RTTOV evolution. Therefore, when we compared the EUMETSAT temperature profiles with ERA5, the IASI version 4 and 5 temperature profiles were linearly interpolated to the 101 pressure levels of IASI version 6.

Table 2 lists the main updates of the L2 algorithm.

**Table 2.** List of IASI L2 updates.

| Date | L2 update | IASI L2 PPF Software Version |
|---|---|---|
| 27 November 2007 | Initial release of IASI/Metop-A L2 | 4.0 |
| 29 April 2008 | Major changes in cloud coverage, surface temperature, and temperature profiles. | 4.2 |
| 21 January 2009 | Surface temperature only provided for cloud-free observations | 4.3.2 |
| 14 September 2010 | Improved T profiles, but available for fewer observations. From this version onwards, temperature profiles and surface temperatures are provided for the same observations. Increased number of cloud-free observations | 5.0.6 |
| 2 December 2010 | Temperature information is now also provided for cloudy pixels (more than half of the IASI observations now have this info.). | 5.1 |
| 20 October 2011 | Improved cloud screening for temperature retrieval. Changed radiative transfer model to RTTOV-10. | 5.2.1 |
| 28 February 2012 | Major change in the cloud detection algorithm, more stringent, resulting in a decrease of the number of cloud-free observations. Temperature information is now provided for observations with a cloud coverage below 25% | |
| 8 March 2013 | Initial release of IASI/Metop-B L2 | |
| 30 September 2014 | Major update in the processing algorithm, improved all-sky retrievals with statistical algorithm upgrades, and synergistic exploitation of IASI with AMSU/MHS. Provision of full retrieval error covariance after OEM. Simplified cloudiness summary | 6.0.5 |
| 24 September 2015 | Updates to the surface temperature algorithms, biases in land surface temperature reduced. | 6.1.1 |
| 2 June 2016 | PWLR exploits information in adjacent channels and becomes PWLR$^3$. + finer atmospheric clustering. Important improvements to the temperature retrieval algorithms | 6.2.2 |
| 20 June 2017 | Enhancements to other IASI L2 products and auxiliary information (CO, SST flags, dust flags ... ) | 6.3.2 |
| 11 April 2018 | Updated $CO_2$ assumptions to contemporary values in OEM, bias in tropospheric temperature reduced. | 6.4 |

## 2.2. ERA5 Temperature Product

ERA5 is the latest ECMWF reanalysis [24,32,33]. ERA5 provides hourly estimates of a large number of atmospheric, land, and oceanic climate variables. It is produced using 4D-Var data assimilation as part of the ECMWF Integrated Forecast System (IFS), with 137 hybrid sigma/pressure levels from

the surface up to 0.01 hPa (80 km). Atmospheric data, such as the temperature profile, are available and interpolated at 37 pressure levels. The IFS is coupled to a land and ocean model, providing the temperature at the surface (e.g., skin and sea surface temperature [34,35]). ERA5 assimilates high spectral resolution infrared radiances from the IASI-A and IASI-B instruments, AIRS on Aqua, and CrIs from S-NPP and NOAA-20, with IASI being the most significant contributor to error reduction for global NWP in the infrared region [36].

ERA5 hourly temperatures are given on a 0.25° × 0.25° latitude-longitude grid. For the comparison with IASI temperatures, ERA5 temperatures were linearly interpolated to the latitudes, longitudes, and time of IASI observations. The temperature profiles were interpolated to the pressure levels of IASI version 6. Interpolating IASI temperatures to ERA5 pressure levels does not significantly change the results.

## 3. Results

L1C and L2 updates are systematically shown in the figures with vertical dashed lines. Depending on whether L1C or L2 is discussed, only the corresponding updates are shown.

### 3.1. Comparison of Operational and Reprocessed L1C Radiance Values

The differences between the reprocessed and operational radiance values were computed at four selected wavenumbers (700, 1200, 1700, and 2200 cm$^{-1}$) and for four 1° × 1° regions with various latitudes/longitudes and characteristics (forest, ocean, land, and ice):

- Amazon rainforest (AMA): 3° S to 2° S, 65° W to 64° W;
- Indian Ocean (OCE): 28° S to 27° S, 72° E to 73° E;
- Western Europe (EUR): 47° N to 48° N, 5° E to 6° E;
- Greenland (GRO): 72° N to 73° N, 43° W to 42° E.

The differences were computed with respect to the Field of Regard (FoR), which ranges from 1 to 30 (1 at the beginning of the scan, 15–16 around the nadir, and 30 at the end of the scan). Each FoR contains four Fields of View (FoV) called pixels.

The observations (date, latitude, longitude, FoR, and four radiance values at 700, 1200, 1700, and 2200 cm$^{-1}$) of each region were selected and then compared by FoR. If a daily file did not contain any observation of the region with a pixel in the right FoR, the difference for this day, region, and FoR was set to Not a Number (NaN).

Figure 2 shows the differences between the reprocessed (R$_{reproc}$) and operational radiance values (R$_{oper}$) in Greenland for the eighth FoR, taken as an example, since IASI has more overpasses close to the poles. Other regions and FoR (not shown here) exhibit similar differences.

Separating day and night observations does not change the result of the comparison and therefore, we mixed both day and night observations in this study.

The differences are all factors of 10$^{-7}$ (at 700 and 1200 cm$^{-1}$) or 10$^{-8}$ (at 1700 and 2200 cm$^{-1}$), which means that the differences have the same order of magnitude as the precision of the radiance values. At all wavenumbers, the differences are larger before 2014. After 2014, there are still a few differences, but they are less frequent and not as large.

In order to check the differences of radiances as a function of FoR, monthly means were computed for each FoR group. Standard deviations of the differences (not shown) have more or less the same value as the mean and the same general evolution. Figure 3 shows the mean of the differences for all FoR groups in the four regions. The differences are shown as percentages of the reprocessed radiance values.

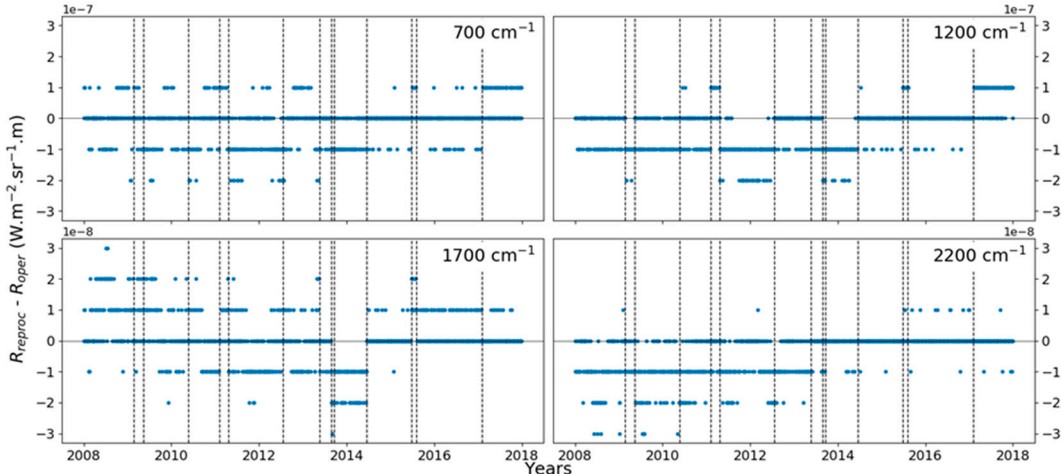

**Figure 2.** Example of differences between the reprocessed and operational radiance values in Greenland at 700, 1200, 1700, and 2200 cm⁻¹ for the eighth FoR, as an example. Vertical dashed lines correspond to L1C updates (see Table 1).

Several changes in the differences appear in the four regions and at the four wavenumbers (they are not visible at 2200 cm⁻¹ because the color scale is too large, but reducing it makes the changes visible):

- A slight increase (at 700 and 1200 cm⁻¹) or decrease (at 1700 cm⁻¹) of positive differences in February 2017. After an investigation, we discovered that these differences are associated with the version 7.4 update (routine scan mirror reflectivity update);
- Positive differences in January 2011–April 2011 and in June 2015–July 2015 at all FoRs;
- Negative differences in May 2011–July 2012 and in August 2013–June 2014. The differences are larger at small FoRs.

At 1200 cm⁻¹, the differences are much larger in Greenland than in other regions. This is due to the fact that outgoing infrared radiance values are smaller over cold surfaces. Furthermore, a main contributor to the differences between operational and reprocessed radiance values at 1200 cm⁻¹ is the scan mirror reflectivity correction used for L1 processing (Table 1). This correction has a higher impact for small FoRs, between 1000 and 1200 cm⁻¹, and for very hot or very cold scene temperatures, as is the case for Greenland.

Other changes are visible at only one wavenumber:

- At 700 cm⁻¹: Large positive and negative differences (± 0.03%, negative at both ends of the scan, positive in the middle) until 2013 in the tropical regions (AMA + OCE). In Greenland and to a lesser extent in Europe, there are seasonal variations of negative and positive differences. The differences decrease after 2013 because of a second change of the Instrument Point Spread Function (IPSF) (see Table 1);
- At 1200 cm⁻¹: Small negative differences (~0.02%) until a decrease in 2010 (improvement of the spectral calibration). Very small differences after 2010. At this wavenumber, differences are larger in Greenland (~0.05%) than in the other regions;
- At 1700 cm⁻¹: Mostly positive differences before 2013 (~0.2%). After 2013, differences are still positive, but a lot smaller. There are seasonal variations in Greenland between 2010 and 2013;
- At 2200 cm⁻¹: Large negative differences before 2013 (~0.5%), and very small differences afterwards.

In the four regions studied, the surface type does not seem to have an impact on the differences. However, the differences are influenced by the latitude (seasonal variations of the differences at high latitudes at 700 and 1700 cm⁻¹).

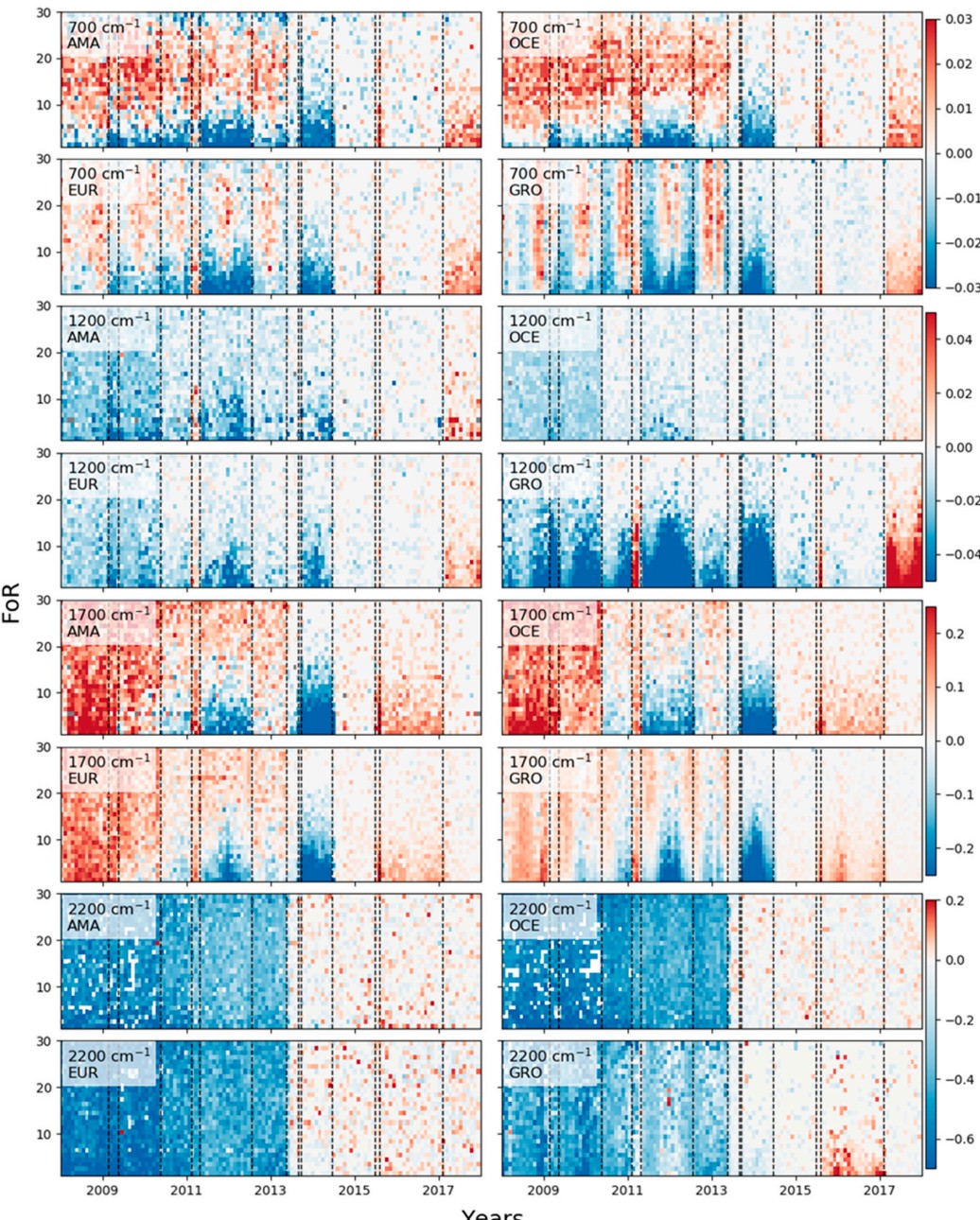

**Figure 3.** Evolution of the differences $R_{reproc}-R_{oper}$ as a function of the Field of Regard (FoR) in the Amazon rainforest (AMA), Western Europe (EUR), the Indian Ocean (OCE), and Greenland (GRO) at 700, 1200, 1700, and 2200 cm$^{-1}$. The differences are shown as a percentage of the reprocessed L1C. Gray pixels correspond to no data with the right FoR. The vertical dashed lines correspond to L1 processing updates. Note that the color bar limits are different for each wavenumber.

In terms of the brightness temperature, the L1C differences correspond to differences of ~0.015 K at 700 and 1200 cm$^{-1}$, ~0.04 K at 1700 cm$^{-1}$, and ~0.1 K at 2200 cm$^{-1}$ when the differences are the largest.

The sign of the differences can vary between one channel and its neighboring channel, but the absolute value does not change.

### 3.2. Comparison of L2 IASI and ERA5 Temperatures

The differences between IASI and ERA5 temperatures were computed at several pressure levels and averaged in 30° latitude bands (all of the daily observations were taken into account for the

computation of the mean). IASI radiance values are assimilated in ERA5, but its assimilation system uses many other satellite data in the same and other spectral ranges, as well as ground-based observational data that it adjusts to, making the ERA5 and IASI temperatures relatively independent.

In Figures 4–7, those differences are plotted in orange for latitude bands in the Northern Hemisphere and in blue for those in the Southern Hemisphere for both Metop A (in dark colors) and Metop B (in lighter colors).

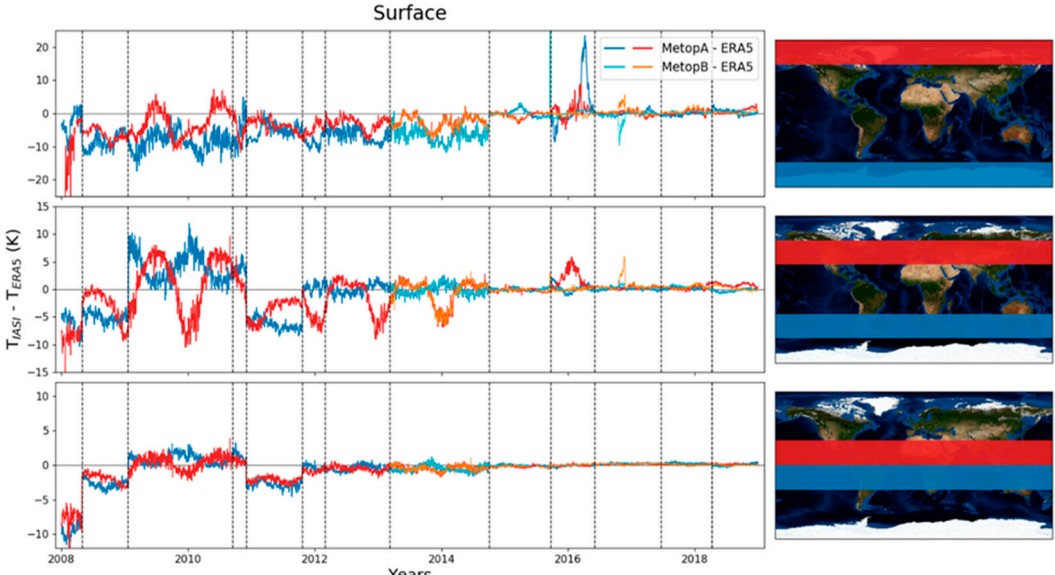

**Figure 4.** Differences between IASI-A and ERA5 (in dark colors), and IASI-B and ERA5 (in lighter colors) for surface temperature at the poles, mid latitudes, and equator. Differences in the Northern Hemisphere are plotted in red/orange and differences in the Southern Hemisphere are plotted in blue. The vertical dashed lines correspond to L2 updates. Note that the y-axis limits for each latitude band are different.

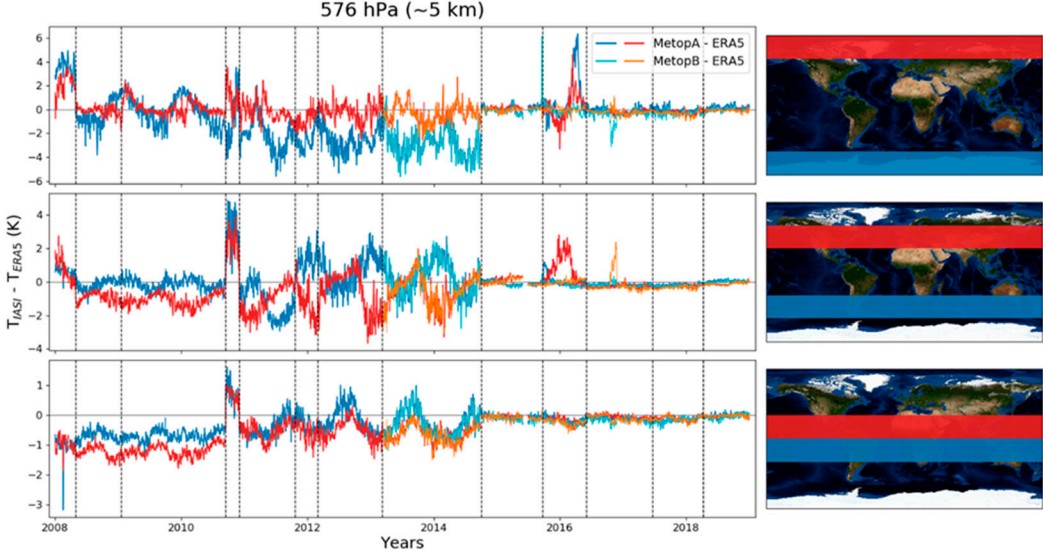

**Figure 5.** Differences between the IASI and ERA5 temperature at 576 hPa for Metop-A and Metop-B at the poles, mid latitudes, and equator. Note that the y-axis limit is different for the different latitude bands. The vertical dashed lines correspond to L2 updates.

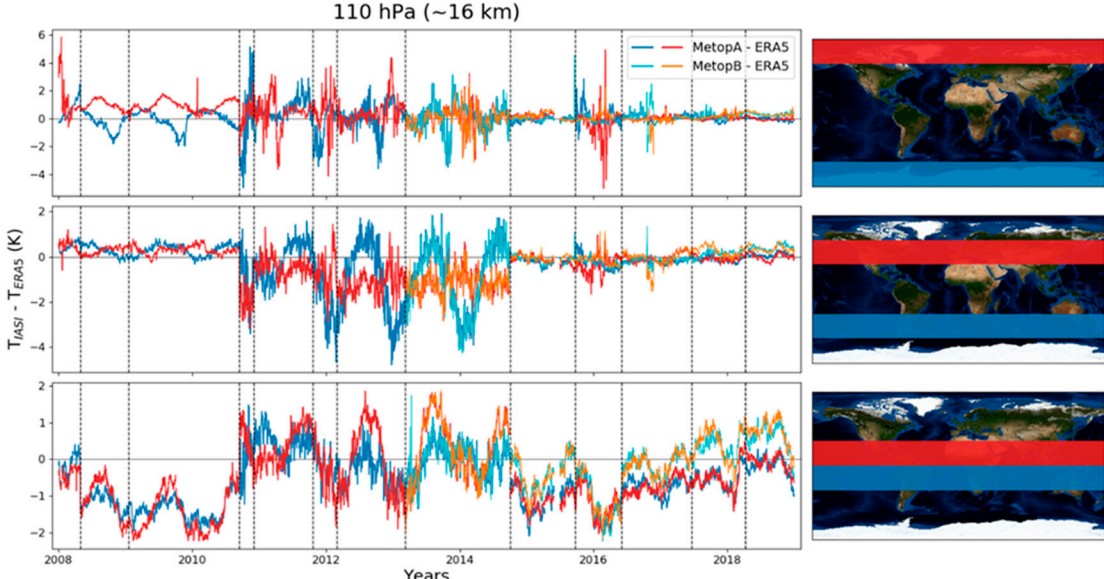

**Figure 6.** Differences between the IASI and ERA5 temperature at 110 hPa for Metop-A and Metop-B at the poles, mid latitudes, and equator. Note that the y-axis limit is different for the different latitude bands. The vertical dashed lines correspond to L2 updates.

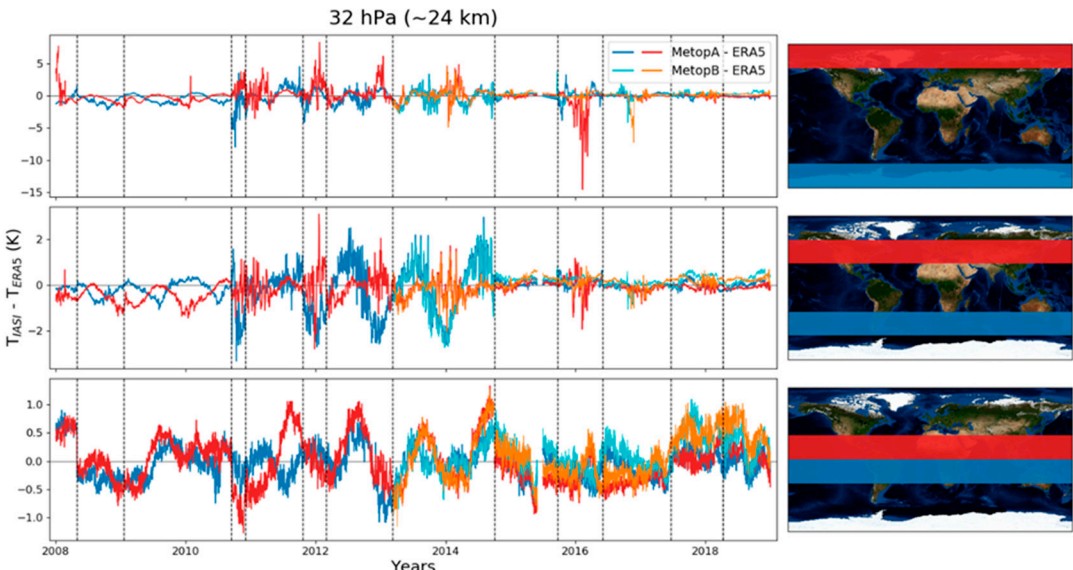

**Figure 7.** Differences between the IASI and ERA5 temperature at 32 hPa for Metop-A and Metop-B at the poles, mid latitudes, and equator. Note that the y-axis limit is different for the different latitude bands. The vertical dashed lines correspond to L2 updates.

At the surface (Figure 4), differences are very large at the beginning of the IASI time series, especially for latitudes larger than 80° N. After an update in 2008 (major changes in cloud coverage, surface temperature, and temperature profiles), the differences decrease at the North Pole (~10 K after 2008) and at the equator (less than 5 K), but they are still quite large. In October 2011, improved cloud screening improves temperature retrievals and differences decrease significantly in the Tropics and mid-latitudes (with seasonal variations in the Northern Hemisphere). In September 2014, the differences decrease again and they are very small afterwards. The change corresponds to a major update in the processing algorithm with the arrival of a new IASI L2 processor. There are a few months in late 2015/early 2016 where the differences between ERA5 and IASI-A increase at mid-latitudes and the

poles. In May 2016, an important improvement to the temperature retrieval algorithms reduces the large differences observed. The same thing happens for IASI-B temperatures in late 2016.

Comparing land temperatures and sea temperatures separately (not shown here) shows larger differences and seasonal variations over land (positive differences in summer and slightly negative differences in winter). However, the general pattern and temperature difference "jumps" seen in Figure 4 are also detected over both land and sea.

To check how IASI compares to ERA5 at different altitudes in the atmosphere, in Figures 5–7, we show the differences between IASI-A and B and ERA5 at 5, 16, and 24 km. In general, differences in temperature are smaller in the atmosphere than at the surface, since at the surface, land properties (emissivity, water content, land use, etc.) play a role in determining Earth's skin temperature and diurnal and seasonal variations in the solar zenith/elevation angle are more important.

At 5 km (Figure 5), there are differences of the order of 1 to 2 K on average at mid-latitudes and the equator until September 2014. At the poles, the differences are larger (~4 K). At mid-latitudes, larger differences are observed between September and December 2010, due to updates related to cloudy pixels. After the update of September 2014, the differences are much smaller, except in early 2016 for IASI-A and in late 2016 for IASI-B at the mid-latitudes and poles. The larger deviation with IASI-B at the end of 2016 is related to the degradation and loss of AMSU channel 15. In the absence of fully valid microwave data, the retrievals were only performed in IR (i.e., with IASI only). The product quality and yield were recovered after reconfiguration of the MW+IR mode, excluding AMSU channel 15 of Metop-B.

At 16 km (Figure 6), the differences are quite small until 2010 and they do not vary much at mid-latitudes and the poles. During the same period, differences are larger at the equator (~1 or 2 K). Between 2010 and 2014, differences are larger at all latitudes and there are important seasonal variations of the differences, especially at the equator (positive differences in July-August, negative in January-February) and between 30° S and 60° S (negative differences in summer and positive in winter). After the update of September 2014, differences are a lot smaller (except at the beginning of 2016) at mid-latitudes and the poles. At the equator, the differences are still quite large after 2014. This may be due to the fact that the vertical precision of ERA5 temperature profiles is not high enough to accurately describe the change of the temperature gradient at the tropopause.

There are also slight differences between IASI-A and IASI-B temperatures after 2017 at mid-latitudes and after 2014 at the equator (~0.5 at midlatitudes and ~1 K at the equator).

At 24 km (Figure 7), differences are quite small (~1 K) until 2010 at mid-latitudes and the poles. Between 2010 and 2014, they are larger, with seasonal variations, and after 2014, the differences are very small (except 2016). At the equator, the differences are small during the whole 2008–2018 period and the algorithm updates do not seem to have an effect on how they change.

In general, the differences between ERA5 and IASI are larger at the poles than at lower latitudes. This may be due to the presence of ice at the poles and, as such, emissivity problems, especially when interpolating icy pixels with non-icy ones.

Figures 5–7 clearly demonstrate the effect of the changes introduced by the continuous improvement of the operational retrieval algorithm on the retrieved temperature profiles. For any further climate analyses of the IASI L2 temperatures, reprocessing of the operational data is mandatory.

## 4. Discussion

Relatively little has been done to generate systematic homogenous records for climate variables with IASI, although the spectral signature of climate variability has been studied for similar instruments (e.g., AIRS [4,37]). The Level 1C radiance values have recently been reprocessed with the latest version of the algorithm by EUMETSAT, but a consistent reprocessing of the Level 2 temperature and humidity series from IASI has not yet been released. This complicates the construction of a homogeneous temperature data record from IASI. This work aimed to show the large, yet undocumented, effects of the different updates that have taken place for both radiance values and temperature.

We note that ERA5 assimilates IASI L1C operational radiance values. This means that the ERA5 temperatures are impacted by the non-homogeneity in the radiance values record. The ECMWF bias correction system [38] and assimilation of data from other instruments significantly reduce this impact, but do not completely eliminate it. This issue is not taken into account in the comparison of IASI L2 and ERA5 temperatures shown in this work, but the order of magnitude of the temperatures ($10^2$ K) is much larger than the L1C changes (up to 0.1 K), so it is assumed that it is not impacted by it.

Clearly, the archived temperature record of IASI as it is now does not allow the construction of climate trends, and care should be taken when using temperature data as an input in algorithms for trace gas concentration retrieval, for which the temperature is needed [22].

## 5. Conclusions

In this study, we first compared IASI's reprocessed radiance values with the operational ones, and showed that there are differences of 0.02% in the reprocessed radiance values at 700 and 1200 cm$^{-1}$, 0.2% at 1700 cm$^{-1}$, and 0.5% at 2200 cm$^{-1}$. In terms of the brightness temperature, these differences correspond to 0.02, 0.04, and 0.1 K, respectively. In all of the regions studied, these differences are found to decrease after two updates from EUMETSAT: At 700 and 2200 cm$^{-1}$, the 2013 change of IPSF has the most impact on the evolution of the differences. At 1200 and 1700 cm$^{-1}$, most of the decrease is due to the 2010 improvement of the spectral calibration. After February 2017, we identified differences at 700 and 1200 cm$^{-1}$ between the operational and the reprocessed dataset that originated in a reprocessing configuration error. Until the reprocessed record is consolidated through 2017, L1C users of IASI-A are recommended to use the reprocessed radiance values from October 2007 to January 2017 and then the operational radiance values from February 2017 onwards.

We then compared the currently available temperature record from the operational EUMETSAT release for IASI-A and B with that of ERA5. We showed that two algorithm updates in December 2010 (changes in the cloud processing) and September 2014 (new IASI L2 processor) have had the largest impact on the temperatures. After 2014, the differences between IASI and ERA5 temperatures were small.

All of the lessons learned during the lifetime of the IASI mission will also help researchers to better prepare and exploit its successor—IASI-NG [39,40]—which will be launched on the Metop-SG program after 2023. Reprocessing IASI L2 data will allow us to have a 35-year homogeneous time series, with IASI and IASI-NG combined.

**Author Contributions:** M.B. performed the comparison and wrote the article with comments from the co-authors. S.S. wrote Section 2.2. J.H.-L. provided information about L1C data and S.W. and L.C. provided the reader for L1C data. M.D.-B., D.C. and E.J. helped understand L1C changes. T.A. provided information about how L2 are computed and helped interpret the jumps in data. This work was supervised by C.C. and S.S. All authors have read and agreed to the published version of the manuscript.

**Funding:** This project has received funding from the European Research Council (ERC) under the European Union's Horizon 2020 and innovation programme (grant agreement No 742909, IASI-FT advanced ERC grant). The authors acknowledge the Aeris data infrastructure (https://www.aeris-data.fr/) for providing access to the IASI Level 1C data and Level 2 temperature data used in this study.

**Conflicts of Interest:** The authors declare no conflict of interest.

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
