# Peer review of "Ten-Year Assessment of IASI Radiance and Temperature"

_remotesensing, doi:10.3390/rs12152393_

Round 1

Reviewer 1 Report

Overall this is a well-written paper describing the radiance measurement implications of the reprocessed IASI-A 2007-2017 radiance dataset (at 4 wavenumbers), and the resulting retrieved temperature differences as compared to ERA5 reanalysis data.  One of the significant conclusions is that some of the IASI inhomogeneous temperature products should not be used for trend analysis.  This is important and should be published.

The manuscript does need some minor changes/adds and verb tense fixes before publication.  Please see below.

- line 60:  "...instruments are required..."

- line 113:  vary should be varies

- line 176: "the version 6 implements a higher level of sophistication,"

- line 177: "...parameters are searched..."

- lines 184-186:  Why not use the lower vertical resolution (but interpolated) AMSU temps instead of a first guess?  Is it because the AMSU data are only avalaible from 2014 on?

- line 269:  add an "a" between "still" and "few"

- line 286:  Why are the differences much larger in Greenland at 1200 cm-1?  Is it because the radiance signal emanating from the cold icecap is so small that even small differences become relatively large?  Seems like a worthwhile geophysical radiative transfer question is avoided here.

- line 314:  at the end of the line, "use" should be "uses"

- Figure 3: since the colorbar limits are different for each wavenumber, why isn't the colorbar range for 2200 cm-1 optimized so the colors show more variability?

- line 381: remove the spurrious << and >>, and "improvements" should be singular

- Figures 4-7:  the differences between the light and dark red/blue lines are not easily discernible.  Consider solid versus dashed lines

- line 415:  "...and diurnal and seasonal variations in solar zenith/elevation angle are more important."

- line 510:  Figure 7 is mislabeled as Figure 6 

- line 520: "...of 0.02% in the..."

- line 527:  algorithms should be singular

- line 524:  at the end of the line, the "do" should be "does"

Reviewer 2 Report

Dear Authors,

please find my comments on your manuscript. My biggest remark is on the lack of reference to other studies performed on this topic (when discussing the results).

Round 2

Reviewer 2 Report

Dear Authors,

I'm very happy with the changes incorporated into the manuscript. I believe that in the current form it has been improved a lot.